# Enhanced biodiversity of gut flora and feed efficiency in pond cultured tilapia under reduced frequency feeding strategies

Scott A. Salger[1], Jimi Reza[2], Courtney A. Deck[1], Md. Abdul Wahab[2,3], David A. Baltzegar[1,4], Alexander T. Murr[1], Russell J. Borski[1]*

1 Department of Biological Sciences, North Carolina State University, Raleigh, North Carolina, United States of America, 2 Department of Fisheries Management, Bangladesh Agricultural University, Mymensingh, Bangladesh, 3 WorldFish, Dhaka, Bangladesh, 4 Genomic Sciences Laboratory, North Carolina State University, Raleigh, North Carolina, United States of America

* russell_borski@ncsu.edu

**Data Availability Statement:** All Prokaryotic and Eukaryotic microbiome Fastq files can be accessed in the National Center for Biotechnology information (NCBI) under GenBank BioProject

## Abstract

Feed constitutes 50–70% of total production costs of tilapia, one of the most widely cultured finfishes in the world. We evaluated reduced-feeding strategies for improving production efficiency of Nile tilapia (*Oreochromis niloticus*). In a 12-week pond trial, fish were fed daily, every other day, every third day, or not at all. Ponds were fertilized to enhance natural foods. In a fifth group fish were fed daily without pond fertilization. Fish fed daily with or without pond fertilization and fish fed every other day had higher specific growth rates, survivability, and net production than the other two treatments. Fish feed efficiency and benefit to cost ratio was highest for treatments fed in a pulsatile manner (i.e. fed every other day or every third day) with fish fed on alternate days providing the best net return among all groups. Fish fed on alternate days had more moderate gene expression levels of intestinal nutrient transporters which may allow for a more balanced and efficient nutrient uptake. Fecal microbe analyses identified 145 families of prokaryotic and 132 genera of eukaryotic organisms in tilapia. The highest diversity of prokaryotes was found in fish fed either every other day or daily in fertilized ponds and the highest diversity of eukaryotes was found in fish fed every other day. These studies indicate feeding Nile tilapia on alternate days along with weekly pond fertilization has no deleterious effects on growth, survivability, or production versus daily feeding regimes, but enhances feed efficiency by 76% and provides the greatest net return on investments. Our studies also suggest for the first time that combining alternate-day feeding with pond fertilization produces the greatest microbial biodiversity in the intestine that could contribute to enhanced feed efficiency and overall health of tilapia.

## Introduction

Aquaculture production is one of the fastest expanding agriculture industries in the world with growth rates in excess of 40% [1]. Tilapia and other freshwater species, namely carps, will make up 60% of the fish cultured worldwide by 2025. Global production of farmed Nile tilapia (*Oreochromis niloticus*) has increased exponentially since 1985, with over 3.7 million metric tons

Accession No. PRJNA485292. All other relevant data are within the paper and its Supporting Information files.

**Funding:** This work was supported by the AquaFish Innovation Lab through the United States Agency for International Development/Oregon State University (USAID EPP-A-00-06-00012-00) and by participating institutions, particularly North Carolina State University and Bangladesh Agricultural University and the NSF (IOS-1457040 to RJB). The funders had no role in study design, data collection and analysis, decision to publish, or preparation of the manuscript.

**Competing interests:** The authors have declared that no competing interests exist.

consumed in 2014 [2]. Currently, tilapia and many other fish species are grown predominantly in pond culture under various conditions: intensive culture with high stocking densities, reliance on commercial feeds, and often requiring mechanical aeration of the ponds; extensive culture where fish are stocked at low densities and organic and inorganic fertilizers stimulate pond primary food production with little to no additional feed supplementation; and, most frequently, semi-intensive culture conditions in which fish are stocked at moderate densities and provided supplemental feeds to complement microorganisms produced naturally with pond fertilization, which can effectively quadruple production over extensive culture systems [3–5].

Feed is the major expense in aquaculture, comprising up to 50–70% of total production costs for tilapia. Methods to limit the amount of feed can help reduce overall costs to producers and has major impacts on small-scale farmers that dominate in under-developed countries. Reductions in feed ration by up to 50% by using alternate-day feeding improves the feed efficiency and economic returns of tilapia monocultured at densities of 2–4 fish/m$^2$ with little impact on growth, survival, or production yield [6].

A better understanding of how finfish acquire and utilize nutrient inputs is requisite for future improvements in aquaculture production efficiency. The underlying mechanism explaining how reduced frequency or alternate-day feeding strategies can achieve equivalent production yields with less feed is poorly understood in finfish. Some evidence suggests that during periods of fasting, nutrient uptake efficiency in the intestine is intrinsically enhanced, leading to a more-efficient uptake of nutrients at the next feeding period [7–10]. Thus, fish being fed a daily regime have lower uptake efficiency and do not receive maximal dietary benefit. The increase in nutrient uptake efficiency following periods of decreased nutrient availability has been postulated, in part, to explain the compensatory growth (CG) response observed in many aquaculture species [7–11]. Additionally, reduced feeding may promote foraging on primary food production within the ponds, leading to a more diverse diet (e.g. algae, insect larvae, plankton), enhancing nutrient recycling within the ponds. This enhanced diversity may directly influence intestinal absorption by promoting colonization and growth of microbes that may aid in nutrient utilization and uptake efficiency that are key to growth and health of the fish [12].

The emerging field of metagenomics has substantial implications for sustainable aquaculture, as diet, feeding strategy, and other environmental factors strongly influence the diversity and constitutive abundance of intestinal microbiota in both humans and fish [12–15]. In finfish, research has shown that probiotic maintenance of beneficial gut flora can promote growth, greater nutrient availability, and better stock health [16–17]. Early studies in channel catfish (*Ictalurus punctatus*) and carp (Cyprinidae) identified several nutrients (e.g. biotin, pantothenic acid, vitamin B12), which are produced by intestinal microbes, but may be limiting in lesser-quality feeds [18–19]. Interestingly, proper intestinal flora in tilapia may also positively impact human health as natural flora inoculates could theoretically out-compete non-natural pathogenic microbes. In Nile tilapia cultured in fertilized ponds found to be contaminated with fecal coliform bacteria in Bangladesh, *E. coli* comprised up to 10% of fecal coliform bacteria identified by classic plating techniques, which could be passed on to consumers through improper storage and handling practices [20].

Absorption of nutrients from feed is facilitated through membrane bound transporters located within enterocytes of the gastrointestinal lumen [21–22]. Nutrient transporters are solute-linked nutrient carriers with roles in transporting and exchanging nutrients, such as amino acids and sugars, and ions across epithelial cellular membranes [23]. Upregulation of nutrient transporter gene expression has been associated with enhanced utilization efficiencies of nutrients from the gut [24] and could play an important role in increasing uptake in tilapia fed at reduced frequencies [6, 25]. Deficiencies in key nutrients may also lead to upregulation of nutrient transporter gene expression or activity resulting in a greater ability to utilize the

limited amount of nutrients available [26]. This could play a role in increased feed efficiency in animals initially deprived of specific nutrients or of tilapia fed at reduced frequencies.

Here, we evaluated potential underlying mechanisms behind the greater feed efficiency in fish fed on alternate days. It is currently unknown as to how reduced frequency feeding strategies may alter the gastrointestinal microbiome and nutrient transporter gene expression of tilapia resulting in better nutrient utilization and feed efficiencies. Here, we evaluated the practices of alternate-day feeding to both establish whether this practice is effective in other areas of the world and to determine if changes in the tilapia gastrointestinal microbiome and nutrient transporter gene expression might account for improved feed efficiency under the management practice. We also sought to assess if reducing the frequency further to every third day might further improve production efficiency of tilapia. We found no difference in growth parameters or survival in tilapia fed on alternate days versus those fed daily. Nile tilapia fed on alternate days had improved feed efficiency and a greater diversity of gut prokaryotic and eukaryotic microorganisms, while transcript levels of select nutrient transporters changed little in fish fed on alternate days versus those fed daily, every third day or not at all. Taken together, this is the first study to suggest that enhanced gut microbial diversity may contribute to improved feed utilization under reduced frequency feeding strategies in the culture of Nile tilapia and perhaps other finfish.

## Materials and methods

All procedures were approved by the Institutional Animal Care and Use Committee (IACUC) at North Carolina State University (approval number: 13-139-0). All animals were euthanized using buffered tricaine methanesulfonate (MS-222) followed by cervical transection following standard procedures.

### Fish and pond sampling

The tilapia growth trial was performed at the Fisheries Field Laboratory, Bangladesh Agricultural University, Mymensingh, Bangladesh. All-male sex reversed Nile tilapia (~3.5 g) were stocked at 5 fish/m$^2$ in 16 ponds (0.1 ha; 4 replicates per treatment), with weekly pond fertilization at a rate of 28 kg N and 7 kg P/ha/week for all treatment groups. Twenty ponds were randomly divided into one of five treatments (4 ponds per treatment). The treatments were: (T1) commercial diet feeding daily with weekly pond fertilization, (T2) commercial diet feeding every other day with weekly pond fertilization, (T3) commercial diet feeding every third day with weekly pond fertilization, (T4) weekly pond fertilization with no supplemental feeding of commercial diet, and (T5) daily feeding with commercial diet with no pond fertilization. Fish were fed with formulated feed (CP Bangladesh, 30% crude protein) initially at 10% and then down to 3% body weight/day based on a standard tilapia feed schedule.

Growth (length and weight) was monitored at 2-week intervals by subsampling of fish over the twelve-week growing trial. Feed rates were adjusted accordingly based on this biweekly random sampling of 50 tilapia from each pond. After approximately 12 weeks (day 114 of the trial), all fish were harvested by seining the fish from the ponds. All fish were collected, counted, and weights and lengths taken to assess final weights and lengths, survival rate and total production. Samples of tilapia anterior intestinal tissue and fecal material from the colon were collected for analysis of nutrient transporter gene expression and gut prokaryote and eukaryote fauna at North Carolina State University, Raleigh, NC (NCSU; see below). Five fish were randomly collected from each pond, euthanized with buffered tricaine methanesulfonate (MS-222) and cervical transection for gene expression analysis. These samples were taken from a 1 cm section of the anterior intestine 5 cm posterior to the duodenal bulb (stomach sphincter where intestine meets the stomach) and placed in RNALater (Thermo Fisher

Scientific, Waltham, MA, USA) at room temperature for shipment to NCSU and subsequently stored at -80˚C until total RNA extraction. All five samples per pond were pooled together prior to total RNA extraction (See Nutrient Transporter Gene Expression, below; n = 4 pooled samples per treatment; N = 20). Tilapia fecal material was collected from the posterior intestine (colon; approximately 5 cm section 1 cm anterior to anus) and rRNA was extracted using a Xpedition Soil/Fecal DNA Miniprep kit (Zymo Research Corp., Irvine, CA, USA) following the manufacturer's protocol (see Ribosomal RNA extraction, below). Samples from 4 fish per pond were pooled together from all 4 replicate ponds for all treatment groups (n = 4 pooled samples per treatment; N = 20). This pooled sample design was used to offset potential variability of microbiota within individuals, instead focusing on common patterns, which may be more reflective of changes with treatment group among the population (pond) as a whole.

## Water quality

Water temperature (˚C), transparency (cm), pH and dissolved oxygen (mg/L) were measured weekly and total alkalinity (mg/L), ammonia-nitrogen (mg/L), nitrate-nitrogen (mg/L), nitrite-nitrogen (mg/L), phosphate-phosphorus (mg/L) and chlorophyll-a (μg/L) were measured biweekly between 0900–1000 h according to previously established protocols [27].

## Collection and quantification of plankton samples

Ten liters of water were collected biweekly from five locations in each pond for plankton sampling. The collected water was filtered through a 45 μm plankton mesh screen. The filtered sample was brought to a standard volume of 50 mL using distilled water. The plankton samples were fixed in 5% neutral buffered formalin in 100 mL bottles for subsequent studies. Plankton was quantified using a 1 mL fixed plankton sample in a Sedgewick-Rafter (S-R) cell counting chamber on a Swift M4000 binocular microscope with phase contrast capabilities. Plankton identifications were made at the Family level.

## Growth and production parameters

Growth and production parameters were calculated as below:

$$\text{Weight gain(g)} = (Mean\ final\ weight\ of\ fish(g)) - (Mean\ initial\ weight\ of\ fish\ (g))$$

$$\text{Specific Growth Rate(SGR; \% body weight/day)} = \left[\frac{Ln(final\ weight) - Ln(initial\ weight)}{Culture\ period\ in\ days}\right] * 100$$

$$\text{Survival rate(\%)} = \left(\frac{No.of\ fish\ harvested}{No.\ of\ fish\ stocked}\right) * 100$$

$$\text{Feed Conversion Ratio(FCR)} = \frac{Total\ feed\ given\ (g)}{Total\ weight\ gain(g)}$$

$$\text{Gross production} = (No.of\ fish\ harvested) * (Final\ weight\ of\ fish\ (g))$$

$$\text{Net production} = (No.\ of\ fish\ harvested) * (Weight\ gain\ of\ fish\ (g))$$

$$\text{Benefit to Cost Ratio(BCR)} = \frac{Total\ revenue}{Total\ cost}$$

## Nutrient transporter gene expression

From the publicly available tilapia genome assembly [28], we identified 6 candidate transporters putatively involved in the digestive transport of amino acids, dietary sugars, and lipids across the intestinal epithelium: (1) facilitated glucose/fructose transporter (*slc2a5*), (2) facilitated glucose transporter (*slc2a6*), (3) long-chain fatty acid transporter (*slc27a4*), (4) Na$^+$-amino acid transporter 2 (*slc38a2*), (5) Na$^+$-amino acid transporter 4 (*slc38a4*), and (6) large neutral amino acid transporter subunit 3 (*slc43a1*).

Tilapia gut transporter mRNA expression was quantified in anterior intestinal tissues (see Fish and Pond Sampling, above, for sampling details) using real-time quantitative PCR (qPCR) performed as previously described with a few modifications [29]. Briefly, total RNA was isolated from the anterior intestine using Tri Reagent (Invitrogen, Carlsbad, CA). One microgram of the total RNA was used to synthesize cDNA using a High Capacity cDNA Synthesis kit (Applied Biosystems, Carlsbad, CA) following treatment with Turbo DNA-free (Ambion, Foster City, CA). RNA was quantified and checked for quality at each step using a Nanodrop ND-1000 spectrophotometer (Thermo Scientific, Wilmington, DE) and agarose gel electrophoresis, respectively. Gene expression (mRNA) of the different transporter forms was measured in the tilapia cDNA using SYBR Green chemistry. Transporter gene-specific primers (S1 Table; Integrated DNA Technologies, Inc., Coralville, IA) were designed using the IDT PrimerQuest Tool (Integrated DNA Technologies, Inc.). Optimization for appropriate annealing temperature, primer concentrations, and cycling parameters was performed using pooled cDNA from the above reverse transcription reactions. One hundred ng of starting total RNA was used for qPCR analysis with Brilliant II QPCR Master Mix (Agilent Technologies, Inc., Clara, CA) containing 1.5 µM gene-specific primers. No template controls and no reverse transcription controls were incorporated into the assay. Pooled cDNA was used to produce the cDNA for creating the standard curves. All qPCR assays were run in triplicate wells on a CFX384 Real-Time PCR System (BioRad Laboratories, Inc., Hercules, CA). Cycling conditions were: 1 cycle, 50°C for 2 min; One cycle, 95°C for 10 min; 40 cycles, 95°C for 15 s, 60°C for 1 min. Melting curve analysis was performed to determine primer specificity. Data was normalized to the starting total RNA concentration. Gene copy number was calculated by comparing the mean cycle threshold (Ct) to the serially diluted cDNA standard curve (R$^2$ = 0.98). The PCR efficiencies were between 90% and 110% for all reactions. The gene expression data were then normalized to the expression of *18S* ribosomal RNA, whose levels were found to be similar across treatment groups.

## Ribosomal RNA extraction

Extraction of ribosomal RNA from tilapia fecal samples (see Fish and Pond Sampling, above, for sampling details) was performed using an Xpedition™ Soil/Fecal DNA MiniPrep Kit (Zymo Research, Corp., Irvine, CA) following the included protocol. Up to 0.25 g of fecal sample was placed into a ZR BashingBead Lysis Tube with 750 µL Xpedition Lysis/Stabilization Solution. The tube was secured in an Xpedition Sample Processor and processed for 30 s and stored at room temperature until extraction. The concentration and quality were determined by

Nanodrop (Thermo Fisher Scientific, Inc., Waltham, MA). The extracted rRNA was stored at -20˚C for sequencing library preparations.

## Prokaryotic 16S and eukaryotic 18S rRNA sequencing library preparation

Prokaryotic 16S and eukaryotic 18S rRNA gene amplicons were prepared following the 16S Metagenomic Sequencing Library Preparation protocol for the Illumina MiSeq system with some modifications. Primers were designed to amplify the V3 to V5 regions of 16S rRNA [30–31] and the V9 region of 18S rRNA [32–33] with overhang adapter sequences compatible with the Illumina index and sequencing adapters. This allowed for double indexing to increase the accuracy of the multiplexed reads (S1 Table). Amplicon PCR was used to amplify the region of interest from the gDNA extracted from the tilapia fecal material samples. A Nile tilapia-specific blocking primer (S1 Table) was introduced during the 18S amplification step to reduce the amplification of tilapia 18S rRNA and increase the amplification of lower abundance eukaryotic rRNA present in the samples. The PCR was performed as follows: 1 cycle of 95˚C for 3 min; 25 cycles each of 95˚C for 30 s, 55˚C for 30 s, and 72˚C for 30 s; 1 cycle of 72˚C for 5 min; and hold at 4˚C. Clean-up of the PCR amplicon products to remove free primers and primer dimers was performed using Agencourt AMPure XP beads (Beckman Coulter, Brea, CA). Fresh 80% ethanol was prepared prior to clean-up. Following amplicon clean-up, index PCR was performed to attach indices to the amplicon PCR products. Dual-index primers were designed so that samples could be multiplexed in one MiSeq lane (S1 Table). Index PCR was performed as follows: 1 cycle of 95˚C for 3 min; 8 cycles each of 95˚C for 30 s, 55˚C for 30 s, and 72˚C for 30 s; 1 cycle of 72˚C for 5 min; and hold at 4˚C. Clean-up of the PCR index products was performed as above. All indexed amplicon concentrations were normalized and amplicons pooled into a single tube. The pooled library was verified for quality and quantified using a High Sensitivity DNA chip on the Agilent 2100 Bioanalyzer (Agilent, Santa Clara, CA). The library was diluted and combined with a PhiX Control library (v3) (Illumina, San Diego, CA) at 10%. The library was sequenced on an Illumina v3 300PE MiSeq run, using standard sequencing protocols at the Genomic Sciences Laboratory at North Carolina State. Base calls were generated on-instrument during the sequencing run using the MiSeq Real Time Analysis (RTA 1.18.54) software and fastq generation. Demultiplexing, adapter trimming, and quality filtering were performed by the MiSeq Reporter Software (2.4 and 2.5.1). The library was run on two lanes to increase the number of reads for each sample.

## Sequence and statistical analyses

The resulting demultiplexed reads were processed using the QIIME 1.9.1 toolkit [34]. Briefly, the paired end reads were joined together and UCLUST [35] was used to search against Greengenes 13_8 reference database [36] for 16S (prokaryote) analysis and SILVA release 119 reference database [37–39] for 18S (eukaryote) analysis, both filtered at 97% identity. Reads not matching a reference sequence were removed from analysis. OTUs were assigned based on a database hit of 97% or greater sequence identity and taxonomy was assigned against the appropriate database. Core diversity analysis was used to perform α-diversity and β-diversity and their rarefaction functions. Weighted and unweighted Unifrac distances [40] were used to compute the between sample diversity which was visualized using principal coordinates analysis (PCoA) plots with Emperor [41].

Significant differences between treatment groups where pond was the experimental unit were determined by one-way ANOVA (Analysis of Variance) followed by Tukey's post-hoc tests or two-way ANOVA analysis followed by pairwise comparisons using t-tests using SPSS (Statistical Package for Social Science) version-16.0, JMP 12 (SAS Institute, Cary, NC),

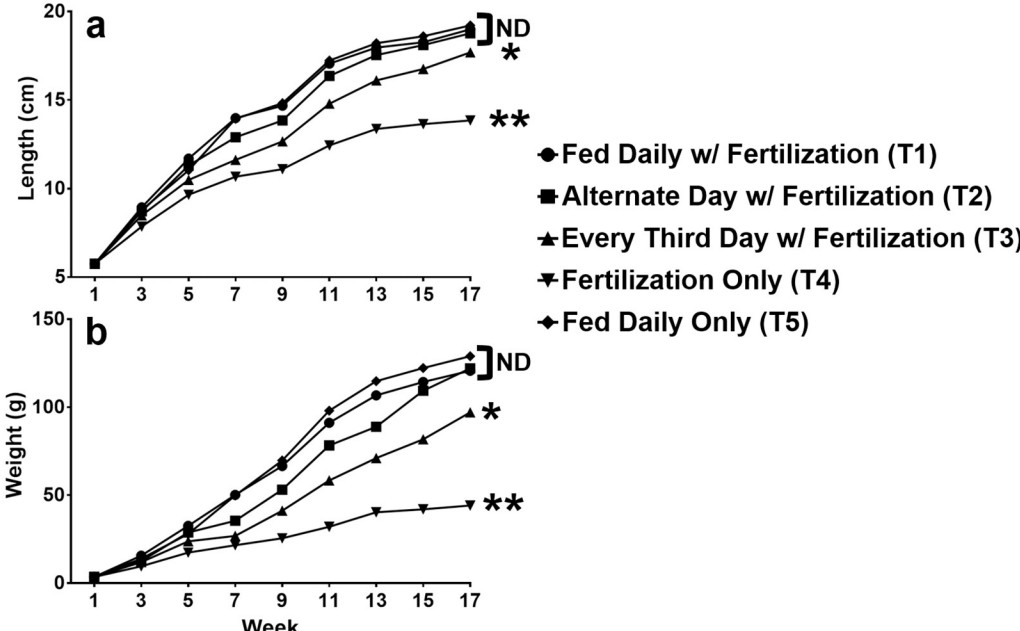

**Fig 1. Mean lengths (a) and weights (b) of Nile tilapia in reduced-feeding regimes throughout the study (error bars have been removed for clarity; see Table 1).** * = $P < 0.05$. ** = $P < 0.01$. ND = No Difference.

GraphPad Prism 6 (GraphPad Software, Inc., La Jolla, CA), or R 3.6.3 [42]. Significance was assigned at an α of 0.05. Venn diagrams were produced using InteractiVenn [43].

## Results

### Effect of feeding frequency on tilapia growth performance and production

There was no significant difference in weight gained, specific growth rate, survival or yield of Nile tilapia between groups fed daily with (T1) or without (T5) pond fertilization and those fed on alternate days with pond fertilization (T2; Fig 1; Table 1). These treatments had the highest fish growth and net production followed by those fish fed every third day in fertilized ponds (T3; Table 1). There were also no significant differences in length (Fig 1a) or weight (Fig 1b)

**Table 1. Growth and production performances of Nile tilapia (*O. niloticus*) subjected to reduced-feeding regimens.**

| | Fed Daily with Fertilization (T1) | Fed Alternate Days with Fertilization (T2) | Fed Every Third Day with Fertilization (T3) | Fertilization Only (T4) | Fed Daily Only (T5) |
|---|---|---|---|---|---|
| Initial Weight (g) | 3.55±0.90 | 3.55±0.90 | 3.55±0.90 | 3.55±0.90 | 3.55±0.90[a] |
| Final Weight (g) | 127.63±2.75[a] | 120.17±5.44[a] | 85.10±11.13[b] | 43.15±4.28[c] | 129.53±8.59[a] |
| Weight Gain (g) | 124.08±2.75[a] | 116.62±5.44[a] | 81.55±11.13[b] | 39.60±4.28[c] | 125.98±8.59[a] |
| SGR (%/day) | 3.14±0.02[a] | 3.09±0.04[a] | 2.78±0.12[b] | 2.19±0.09[c] | 3.15±0.06[a] |
| FCR (Feed Conversion Ratio) | 1.64±0.10[c] | 0.93±0.09[b] | 0.68±0.15[a] | - - | 1.61±0.10[c] |
| Survival Rate (%) | 93.44±6.26[a] | 91.66±8.00[a] | 90.70±9.74[ab] | 76.79±2.68[b] | 97.71±2.11[a] |
| Total Yield (kg/ha) | 6,282.09±354.52[a] | 5,837.84±527.83[a] | 4,179.08±640.07[b] | 1,950.35±233.89[c] | 6,578.53 ±461.41[a] |

Values are mean ± SD. Values with different letters are significantly different ($P < 0.05$).

**Table 2. Economic analyses of Nile tilapia (*O. niloticus*) subjected to reduced-feeding regimens.**

| | Fed Daily with Fertilization (T1) | Fed Alternate Day with Fertilization (T2) | Fed Every Third Day with Fertilization (T3) | Fertilization Only (T4) | Fed Daily Only (T5) |
|---|---|---|---|---|---|
| *Expenditure (USD/pond)**†* | | | | | |
| Fingerlings cost | 11.99±0.65 | 11.99±0.65 | 11.43±2.30 | 11.43±2.30 | 12.64±0.65 |
| Feed cost | 113.88±7.20[b] | 59.82±2.16[c] | 29.87±7.64[d] | -- | 124.05±8.19[a] |
| Lime Cost | 0.58±0.03 | 0.58±0.03 | 0.55±0.11 | 0.55±0.11 | 0.61±0.03 |
| Fertilizers cost | 4.60±0.25 | 4.91±0.27 | 4.67±0.94 | 4.67±0.94 | -- |
| Total Expenditure | 131.05±7.98[a] | 77.29±3.01[b] | 46.52±10.95[c] | 16.65±3.35[d] | 137.30±8.73[a] |
| *Income**†* | | | | | |
| Gross return (USD/ha) | 9,774.19±1,081.58[ab] | 9,082.98±981.20[b] | 5,418.46±659.15[c] | 1,770.14±294.04[d] | 10,235.40 ±1,063.34[a] |
| Net return (USD/ha) | 898.09±563.51[c] | 3,853.63±813.67[a] | 2,250.83±997.06[b] | 610.42±212.27[c] | 1,421.67±612.94[c] |
| BCR (Benefit Cost Ratio) | 1.10±0.06[b] | 1.74±0.16[a] | 1.70±0.35[a] | 1.52±0.18[a] | 1.16±0.07[b] |

Values are mean ± SD. Values with different letters are significantly different ($P < 0.05$).

* USD = United States Dollar; BDT = Bangladeshi Taka. Fish were sold at $1.56 USD (120 BDT)/kg [1 BDT = $0.013 USD].

†Leasing and other operational costs such as labor were excluded from these calculations.

between groups fed daily with (T1) or without (T5) fertilization throughout the trial (two-way ANOVA; $P > 0.05$). There was, however, a significant difference between these two treatments (T1 and T5) and the fish fed on alternate days in fertilized ponds (T2) during weeks 7 through 13 ($P < 0.05$), but no significant difference between these three treatments at weeks 15 and 17 ($P > 0.05$; Fig 1a). Weight followed similar trends, except that there was also no significant difference between these three treatments (T1, T2, and T5) during weeks 13, 15, and 17 ($P > 0.05$; Fig 1b). Fish that were grown in fertilized ponds without supplemental feeding (T4) showed the lowest growth among all groups. The fish produced with pond fertilization alone also had the lowest survival relative to the other groups (Table 1), whose survival rates ranged from 92–98%. The estimated feed conversion ratio (FCR) was lowest for fish fed every third day with pond fertilization (T3), followed by fish fed on alternate days with pond fertilization (T2; Table 1). Fish fed daily had the lowest feed efficiency or highest FCRs regardless of whether ponds were fertilized (T1) or not (T5) as these groups had greater feed inputs, but similar growth to those fed on alternate days with pond fertilization (T2).

The economic assumption of alternate-day feeding is that it would lead to more cost-effective tilapia aquaculture production, which could enhance income of small-scale farmers and consumption of nutrient rich fish. A marginal cost analysis from this study indicates feeding on alternate days along with weekly pond fertilization (T2) gives the best return on investment (Table 2) with much of the cost reduction associated with application of half the feed in alternate day fed fish. Net return values were also highest for fish fed on alternate days with pond fertilization (T2), followed by those fed every third day with pond fertilization (T3; Table 2). Overall, the group fed on alternate days along with weekly pond fertilization (T2) had the highest benefit to cost ratio (BCR) but was not significantly different from groups fed every third day or not fed supplemental feeds in fertilized ponds (T3 or T4); all three of these treatments had significantly higher BCRs than fish fed daily regardless of pond fertilization (T1 and T5).

## Pond water quality and plankton composition

All water quality parameters among groups were well within the range for suitable growth of fish in pond culture (S2 Table) [4, 44]. Ammonia was highest in the group fed daily with pond

fertilization (T1) and lowest in the fertilization alone group (T4). Both nitrites and nitrates were higher in fertilized ponds (T1-T4) and lowest in non-fertilized, fed daily ponds (T5). Dissolved oxygen levels were highest in ponds either fed every 3 days with fertilization or those fertilized only (T3 and T4) and lowest in ponds fed daily independent of whether ponds were fertilized or not (T1 and 5). The pH was highest in the ponds that were fertilized only (T4) and lowest in ponds that were fed a commercial diet (T1-T3, T5).

Six Families of eukaryotic phytoplankton and four groups of zooplankton were identified over the course of the pond trials (S3 Table). The eukaryotic phytoplankton Families identified were Bacillariophyceae, Chlorophyceae, Cyanophyceae, Euglenophyceae, Rhodophyceae, and Xanthophyceae and zooplankton were Copepoda, Crustacea, Rotifera, and Cladocera. T5 had significantly higher counts of the phytoplankton Family Euglenophyceae (4035.7 ± 466.8 SEM cells/L) whereas T1 had the lowest counts of this family (1750.0 ± 252.5 SEM cells/L; $P = 0.0215$). There was no difference in total eukaryotic phytoplankton between treatments ($P = 0.0851$). For zooplankton, T1 had significantly higher counts of Phylum Rotifera (6482.1 ± 1291.0 SEM cells/L; $P < 0.001$) and Order Cladocera (1464.3 ± 198.4 SEM cells/L; $P = 0.0001$) and significantly higher total zooplankton (11571.4 ± 1854.9 SEM cells/L; $P = 0.0003$) than the other treatments. There was no difference in total plankton between any of the treatments ($P = 0.221$).

## Effect of feeding frequency on intestinal nutrient transporter expression

The mRNA levels of six intestinal nutrient transporters were measured via real-time qPCR in the anterior intestine of fish from the growth trial. The nutrient transporters evaluated were: (1) facilitated glucose/fructose transporter (*slc2a5*), (2) facilitated glucose transporter (*slc2a6*), (3) long-chain fatty acid transporter (*slc27a4*), (4) Na+-amino acid transporter 2 (*slc38a2*), (5) Na+-amino acid transporter 4 (*slc38a4*), and (6) large neutral amino acid transporter subunit 3 (*slc43a1*). The efficiency of all real-time qPCR reactions were between 90–110% and the standard curve correlations were greater than 0.97 for all assays performed. Our resulting gene expression values (copy number / ng Total RNA) were normalized to *18S* rRNA levels (no significant difference between treatments was observed in the *18S* rRNA gene).

There was no significant difference in gene expression of the 6 nutrient transporters analyzed between treatments where supplemental food and pond fertilization was provided (T1—T3) regardless of the frequency of feeding, except that *slc38a2* was significantly lower in T2 than T1 and T3 fish (Fig 2). Gene expression for the *slc2a5*, *slc2a6*, *slc27a4*, and *slc38a2* transporters was significantly higher in fish from T4 (pond fertilization only) than T5 groups (daily feeding only). There was no difference among treatments in the mRNA abundance of the *slc38a4* and *slc43a1* transporters (Fig 2e and 2f).

## Effect of feeding frequency on the gut microbial community—prokaryotic assessment

Following quality filtering, a total of 715,725 paired end sequences (330,883 for 16S; 384,842 paired end sequences for 18S) were obtained from the V3—V5 regions of 16S rRNA (prokaryote) and the V9 region of 18S rRNA (eukaryote). All resulting Fastq files can be accessed in the National Center for Biotechnology information (NCBI) under GenBank database BioProject Accession No. PRJNA485292.

A total of 20 prokaryotic phyla, 43 classes, 92 orders, and 145 families were found in the fecal material of tilapia (100% bacteria, no archaea). The dominant phyla identified from these samples belonged to the Fusobacteria (80.4%), Firmicutes (13.8%), Cyanobacteria (2.4%), Bacteroidetes (1.3%), and Proteobacteria (1.3%) (Fig 3a, S4 Table). The proportions of the

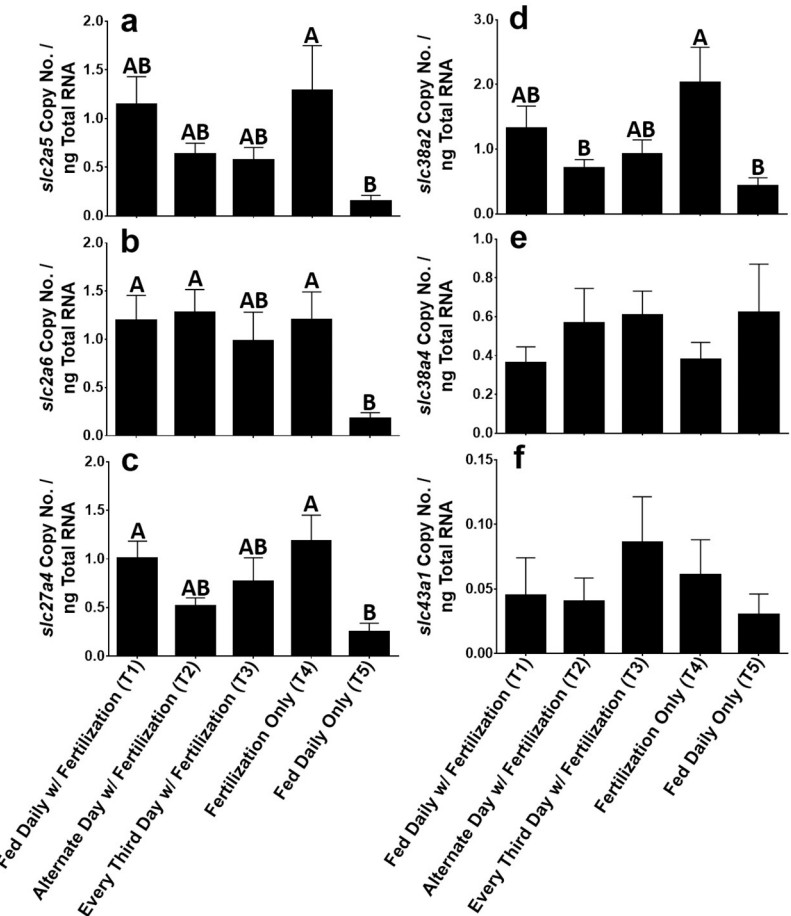

**Fig 2. Gene expression of nutrient transporters in the intestine of Nile tilapia subjected to reduced-feeding regimes (mean ± SEM).** (a) *SLC2A5*; (b) *SLC2A6*; (c) *SLC27A4*; (d) *SLC38A2*; (e) *SLC38A4*; (f) *SLC43A1*. All values reflect copy number / ng Total RNA normalized to *18S* rRNA levels. Treatments with different letters are significantly different (*P* < 0.05).

identified microbes varied between treatments with an increase in the proportion of Fusobacteria (62.6% to 84.6%) and a decline in the proportion of Firmicutes (32.1% to 11.5%) with reductions in feeding frequency from daily to every 3rd day (T1-T3). The fish fed a commercial diet only with no fertilization (T5) had the highest proportion of Fusobacteria (90.8%) and lowest proportion of Firmicutes (6.5%) among the treatments. There were 10 unique bacterial OTUs identified in T1, 6 in T2, 1 in T3, 19 in T4, and 4 in T5 (Fig 4, S5 Table).

Alpha diversity and rarefaction curves were determined for all treatments evaluated. T1 and T2 had the highest Chao 1 diversity index among the groups (*P* < 0.05; Chao1 index mean ± SEM; T1, 392.9 ± 108.5 and T2, 371.8 ± 81.0). The bacterial diversity in the fecal material in tilapia guts declined as the frequency of feeding was reduced to every 3rd day (T3) or where no supplemental feeds were applied (T4) when paired with pond fertilization (Fig 5a, S1a Fig). Although there was no significant difference in species richness due to sample variation within treatments (*P* = 0.256), treatments with greater access or variety of foodstuff (feed and fertilization) had higher diversity measures. The Chao 1 index was highest for T1 and T2. T3 (187.3 ± 77.6), T4 (244.0 ± 44.2), and T5 (215.5 ± 66.4) had similar diversity indices. Our

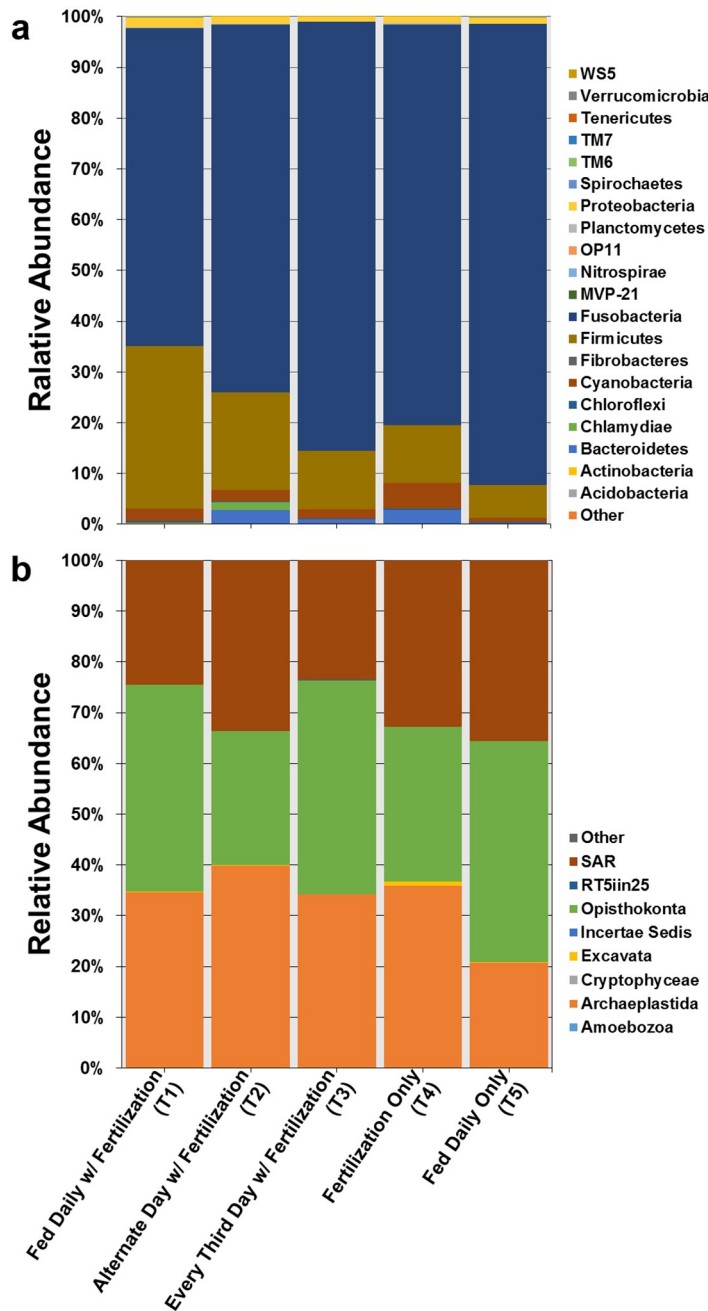

**Fig 3. Relative abundance of prokaryotic (a) and eukaryotic (b) organisms at the level of Phylum in the fecal material of Nile tilapia subjected to reduced-feeding regimes.**

results indicate that there are some community differences between the treatments in our study with PC1 explaining 11.89% of the variation between communities (Fig 6a and 6b). T1 and T5 (both fed daily) had the most similar profiles while T4 (fertilization only) was the most dissimilar. The variation between treatments indicates that there is some distinction between the microbial communities.

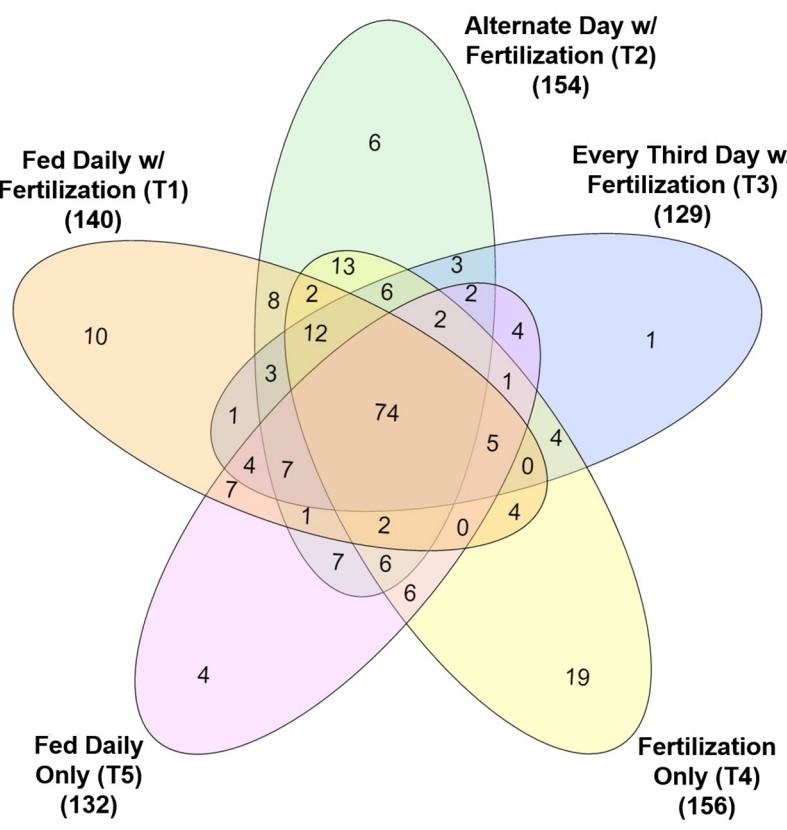

**Fig 4. Venn diagram showing the number of shared and unique Operational Taxonomic Units (OTUs) between each treatment.** Numbers in parenthesis are the total number of OTUs identified in each treatment.

## Effect of feeding frequency on the gut microbial community—eukaryotic assessment

All resulting Fastq files can be accessed in the National Center for Biotechnology information (NCBI) under GenBank database BioProject Accession No. PRJNA485292. A total of 8 major eukaryotic groups and 132 genera were associated with the fecal material of tilapia used in this study. T2 (fed alternate days with pond fertilization) had the greatest number of reads for all groups (58,450) followed by T4 (fertilization only; 49,112), T1 (feed every day with fertilization; 38,378), T5 (feed every day only; 31,758), and T3 (feed every third day with fertilization; 25,752). The dominant groups identified from these samples belonged to the Opisthokonta (Metazoans and Fungi; 41.2%), Archaeplastida (green plants and red algae; 35.4%), and SAR supergroup (Stramenopiles, Alveolates, and Rhizaria, Dinoflagellates, Diatoms, Oomycetes, etc.; 23.3%) (Fig 3b, S4 Table). There were no significant differences in abundance of particular eukaryotic groups among the treatments. Class Chlorophyceae (green algae), Class Mediophycae (diatoms), Phylum Rotifera, and group Magnoliophyta (flowering plants) made up the highest proportion of identified eukaryotic organisms (not shown). T1 and T3 had the highest proportions of rotifers (31% and 35%, respectively). The largest proportions of Magnoliophyta were in T2 (19%) and T3 (20%) and Chlorophycae were in T1 (27%) and T4 (24%), while T5 had the highest proportion of Mediophycae (21%) relative to all other treatments. However, there were no significant differences in abundance of eukaryotes among the treatment groups.

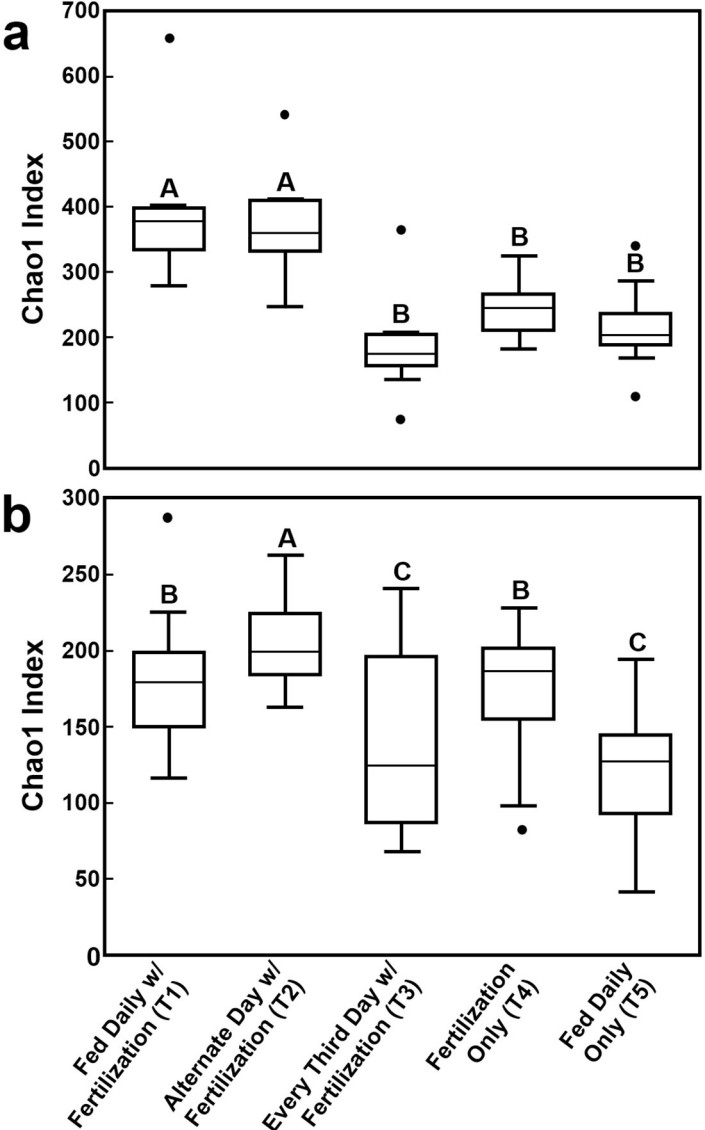

**Fig 5. Chao1 diversity boxplots of microbial communities identified from the anterior intestine of Nile tilapia subjected to reduced-feeding regimes (mean ± SEM).** (a) 16S rRNA (prokaryotes); (b) 18S rRNA (eukaryotes). Treatments with different letters are significantly different ($P < 0.05$).

The most abundant identified groups overall include the rotifers (50,705 counts), green algae (34,755 counts), and flowering plants (19,951 counts).

Alpha and beta diversity measures were also determined for the 18S amplicons. T2 had the highest diversity (Chao1 index, mean ± SEM, 206.9 ± 33.6), followed by T1 (182.0 ± 52.4), T4 (172.0 ± 50.4), and T3 (141.5 ± 59.6) ($P < 0.05$). T5 had the lowest diversity measure at 123.9 ± 47.1 (Fig 5b, S1b Fig). Again, the results indicate that there are some community differences between the treatments in our study with principal component 1 explaining 22.28% of the variation between communities (Fig 6c and 6d). The groupings of these samples, although with overlap, indicate that there are differences in the microbial communities between treatments.

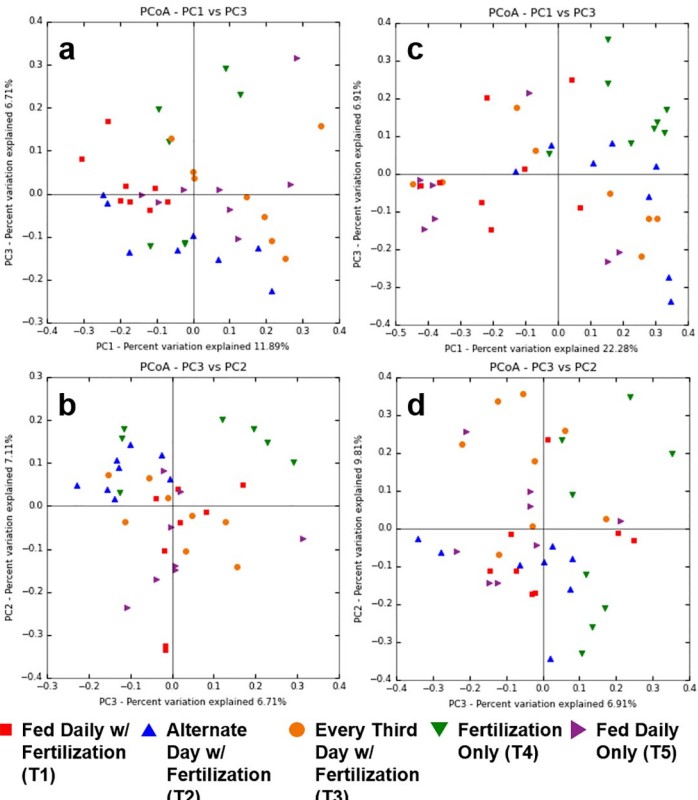

**Fig 6. Principal coordinates analysis (PCoA) plots of prokaryotic (a and b) and eukaryotic (c and d) communities found in the fecal material from Nile tilapia subjected to reduced-feeding regimens.** Each replicate sample is represented in the plot.

## Discussion

It is estimated that 50–70% of total variable costs for growing tilapia and many other fishes is attributable to formulated feeds. Our study demonstrates that alternate-day feeding along with pond fertilization reduces the costs of feeds for growout of Nile tilapia by 50%, increases feed efficiency by 76%, and has little impact on growth, survival, or yield of fish farmed in ponds. This is noteworthy as both ponds fed every day with (T1) or without (T5) pond fertilization had the highest expenditures in our study while ponds that were fertilized only (T4) with only pond fertilization had the lowest cost input. Tilapia producers can see marked reduction in production costs and the highest benefit: cost ratio (BCR) largely due to the reduced feed costs associated with alternate-day feeding strategies. The overall improved capacity of fish fed on alternate days to utilize nutrients for growth (lower feed conversion ratio, FCR) was accompanied by intermediate expression of genes for proximal intestinal solute transporters and elevated diversity of gastrointestinal prokaryotes and eukaryotes, suggesting that these factors along with the higher biodiversity of organisms available for consumption (commercial diet and natural pond productivity) may contribute to enhanced feed efficiency while sustaining similar growth and health of tilapia when fed daily.

The results suggest that the optimal BCR and net return for tilapia production occurs when fish are fed on alternate days in fertilized ponds. This is contrary to studies performed in channel catfish. Fingerling growth, weight at harvest, and FCR were all lower in channel catfish fed on alternate days versus those fed every day [45]. Tilapia are omnivorous, and more so than

catfish, readily feed on diverse food sources throughout their life, including phytoplankton and zooplankton or other natural pond foods enhanced through fertilization or application of supplemental formulated feeds. Hence, tilapia fed on alternate or even every third day likely forage at a greater rate between meals on pond organisms produced through enhanced nitrogen and phosphorous inputs from fertilizers or feeds than fish fed daily. Less reliance on supplemental feeds will increase overall feed efficiency as demonstrated here with low FCR, albeit at a detriment to growth when animals are fed at a frequency of every three days. Clearly, sufficient nutrients, derived in this case from formulated feed, is essential to maximally stimulate growth as we found alternate day fish achieved weights and lengths similar to daily fed fish while those fed every third day did not. The responses to alternate day feeding we observed here with studies in Bangladesh corroborate those previously shown in the Philippines on commercial farms where tilapia were stocked at a 20% lower density and fed a formulated diet composed of high levels of fishmeal and rice bran [6]. Water quality parameters are affected by excess nutrients, but small variations within normal range are unlikely to account for any of the changes in tilapia growth observed in this study. All water quality parameters were well within the suggested range for growing tilapia in ponds (S2 Table) [4, 44]. Collectively, the findings indicate that addition of supplemental feeds can enhance growth of tilapia by 2–4 fold depending on the frequency of feed application and that alternate day feeding is the most cost-effective feed management method for maximizing growth and yield of tilapia in semi-intensive culture, a strategy that is likely to be applicable for pond production of tilapia throughout the world.

Reduced feeding strategies can achieve equivalent production yields with less feed; however, the mechanisms are poorly understood [7–11]. Absorption of nutrients from feed is facilitated through membrane bound transporters located within enterocytes of the gastrointestinal lumen [21–22]. We evaluated if changes in expression of key solute-linked nutrient carriers (SLC), whose abundance may impact nutrient absorption efficiency [21] might be associated with the changes in feed efficiency and growth of Nile tilapia shown here under the different feeding protocols. We used the proximal (anterior) portion of the small intestine to measure solute transporter gene expression levels. Tilapia have a long, coiled intestine in order to process and absorb nutrients from their mainly herbivorous or omnivorous diet. Tilapia lack a pyloric caecum, thus the small intestine is the region of highest nutrient absorption, especially in the proximal area which is the most developed region of the small intestine and exhibits the highest enzymatic activity and absorption of nutrients [46–47]. A number of ATP-binding cassette (ABC) efflux transporter genes associated with multi-drug resistance and efflux of xenobiotics are also expressed in the proximal small intestine of Nile tilapia [48] and SLCs are expressed in a similar region of the intestine in zebrafish (*Danio rerio*) [49].

The expression of solute transporters *slc2a5*, *slc2a6*, *slc27a4*, and *slc38a2* in the proximal intestine of the Nile tilapia were significantly higher in fish grown in ponds where only fertilizers were applied (T4) than in fish grown in ponds which were fed daily without the addition of fertilizers (T5) (Fig 2). The seemingly low expression of all transporters examined in the intestines of fish fed daily without pond fertilization (T5) may in part be the result of receiving similar quantities of feed with the same nutrient profile day in and day out, hence preventing the need for any adaptive response from the intestine [50]. It may also be due to higher rates of intestinal growth in these fish. Increased food consumption can induce lengthening of the intestine and proliferation of the villi to enhance nutrient uptake as well as increase cell turnover rates. This in turn can increase the proportion of immature, non-transporting cells and cause a non-specific change in overall transporter abundance [50]. Conversely, the relatively high expression of these transporters in fish grown in ponds where only fertilizers were applied (T4) likely indicates that gaining nutrition from natural flora and fauna boosted by fertilization

application alone may lack sufficient nutrients necessary to maintain growth and health that would otherwise be available with commercial diets. Thus, gene expression for these transporters may be increased due to the lack of nutrients and for preparing the intestinal cells to rapidly take them up once they become available [50]. Interestingly, fish grown in ponds that received feed and fertilization (T1, T2, and T3), particularly those fed every two (T2) or three days (T3), had intermediate mRNA abundances of certain solute transporters (Fig 2). This suggests that they could be more efficient at nutrient uptake and utilization, particularly when commercial feeds are only periodically available, and that they may perhaps have a greater variety of transporters expressed at lower quantities due to an increased diversity and abundance of natural food sources in the diet. It is interesting that the fish in T1 (daily feeding with fertilization) had approximately 3-fold higher levels of these transporters than T5 (daily feeding only) as both groups were fed daily, but is likely the result of a more varied diet and greater overall abundance of food sources.

The solute transporters *slc2a5*, *slc38a2*, and *slc27a4* exhibited similar expression profiles across the five treatments. When a nutrient yields high metabolizable energy, high concentrations will typically stimulate an upregulation of its transporter as is the case for sugars and fatty acids. By contrast, when a nutrient can be toxic at high concentrations, it will normally cause a reduction in the expression of its transporter [50]. GLUT5, the protein produced from the *slc2a5* gene, is a facilitated fructose transporter in vertebrates [23, 51]. Although a member of the facilitated glucose transporter superfamily (SLC2), glucose appears to be a minor component of its mechanisms. GLUT5 is essential for the transport of fructose across the luminal membrane of the small intestine and transcription of *slc2a5* in the intestine is rapidly increased when a high-fructose diet is consumed [52–54]. Thus, the relatively higher levels of this transporter in fish from ponds with fertilization only (T4) may indicate a greater abundance of fructose in the natural flora and fauna relative to the commercial diet. Similarly, *slc27a4* gene expression was highest in fish grown in fertilized ponds without commercial feed as well as in fish fed at different frequencies along with pond fertilization. These increases may be dictated primarily by the nutrients available in the natural products produced by pond fertilization. *Slc27a4* encodes the FATP4 fatty acid transporter. It promotes lipid absorption in the mouse ileum [55] and the gene is located in a growth-related quantitative trait loci (QTL) in the orange-spotted grouper (*Epinephelus coioides*) [56]. The bacterium *Clostridium ramosum* increases *slc27a4* gene expression with high-fat diets in mice [55]. Although *C. ramosum* was not directly identified in the guts of the Nile tilapia in this study, several sequences did align to members of *Clostridium sp*. OTUs.

Contrary to the fructose and fatty acid transporters, gene expression of *slc38a2* increases in response to the absence of amino acids. This increase leads to the translation of the gene to make the SNAT2 protein which has been shown to aid the recovery of cell volume following amino acid starvation or hypertonic stress [57–61]. In addition, amino acid transporters may serve as sensors of nutrient supplies to regulate expression of the transporters and thus nutrient uptake by the cells [62]. The results show that expression of this gene is upregulated in T4 with only pond fertilization and lower in fed treatments regardless of fertilization. This implies that natural biota increased through fertilization of ponds may lack the full complement of amino acids necessary for the intestine to maintain cellular osmolality. In this case again the *slc38a2* would be upregulated to improve the efficiency of absorbing amino acids under limiting conditions whereas in fish receiving commercial feeds that typically have adequate protein or amino acids, the transporter is expressed at lower levels to prevent excessive amino acid uptake which can lead to toxicity [50, 63].

This investigation also assessed how the gut microbial flora is altered by feeding/fertilization strategies that could potentially identify microbes beneficial to tilapia growth and health. The

establishment of beneficial gut flora to increase nutrient absorption is an emerging research focus in human biology and aquaculture science [17] and may serve to augment existing practices of sustainable feeding and reduction in environmental footprint.

The most abundant phylum of prokaryotes identified in all treatments were the Firmicutes. The Firmicutes are a group of mostly Gram-positive bacteria and are important members of the gut microbiome of many vertebrates. The proportions of Firmicutes identified declined with reduced application rates of commercial feed in fertilized ponds (Fig 3). An increase in the relative abundance of Firmicutes is indicative of obesity in mammals and positively correlated with caloric intake in bony fishes [64–66]. The higher proportion of Firmicute bacteria in tilapia from treatments fed daily in fertilized ponds in our study indicates that there may be an abundance of high-calorie food sources available to these fish. Bacteria from the classes Fusobacteriaceae and Clostridiaceae were most abundant in the fecal material of Nile tilapia used in our study. These groups include many pathogenic strains that could cause disease with immunosuppression or injury to the gut epithelium [67–68]. *Cetobacterium somerae* (a Fusobacteriaceae) had the highest number of reads and proportional abundance in our study with no apparent trend in regulation by feeding or fertilization. *C. somerae* is an indigenous bacterium in the intestinal tract of cultured freshwater fish and is highly efficient at producing Vitamin B12, an essential micronutrient [69–70].

The highest growth and survival rates were observed in T1 (daily feeding along with pond fertilization), T2 (alternate day feeding along with pond fertilization), and T5 (daily feeding only). We identified 7 OTUs (corresponding taxonomic assignments: *Acutodesmus* sp., *Cyanobacterium* sp., *Bacillus* sp., *Blautia* sp., *Anaerovorax* sp., *Sphingomonas* sp., and *Desulfococcus* sp.) that were common between all three of these treatments. The identification of the species relating to these OTUs are currently unknown. Future work will be necessary to determine not only the species but also the exact functions of these microorganisms and to determine whether they might serve to benefit growth of Nile tilapia.

Six families of eukaryotic phytoplankton and four groups of zooplankton were identified from the ponds in this study. Of those identified, plankton from the Families Bacillariophyceae, Chlorophyceae, Euglenophyceae, Rhodophyceae, Subphylum Crustacea, and Phylum Rotifera were also identified in the microbial communities in the fecal material from the Nile tilapia in these studies. Although the microscopic analysis used for pond water only allowed us to identify plankton that were large enough to be viewed, the commonalities with the fecal matter indicate that the tilapia were consuming plankton from the pond as an additional food source. The Holozoans were the dominant group of eukaryotes identified in Treatments 1 (daily feeding along with pond fertilization), 3 (every third day feeding along with pond fertilization), 4 (pond fertilization only), and 5 (daily feeding only) with the most abundant organisms within this group being the Rotifers. As part of the zooplankton community in the ponds, the rotifers make up a large proportion of the food available to the tilapia, outside of commercial feeds. By contrast, T2 (alternate day feeding along with pond fertilization) was dominated by the group Chloroplastida which includes green plants and algae. The Chloroplatids made up a large proportion of eukaryotes identified in all treatments and the most highly identified sequences within this group aligned to *Zea mays*. *Zea mays* could be a component of the feed used in these trials but OTUs were also found in large proportions in the guts of the fish from T4 (pond fertilization only) which did not receive any commercial feed. Cholophytes were also identified as part of the phytoplankton community using microscopic analysis. As such, these OTUs may align to other plant sequences similar to *Zea mays*, that are not found in the available database. The other groups with highest proportions identified across all treatments were the groups Alveolata (ciliated protozoans and dinoflagellates) and Stramenopiles (diatoms and algae). All of these would be components of the natural production in established ponds by

either fertilization or through the process of breakdown of feed and fish waste. Thus, it is apparent that the tilapia in this study were feeding on a wide variety of eukaryotic organisms in the ponds. Being omnivores, tilapia will utilize any food resources available to them for growth and as such a decrease in the application frequency of the commercial feed would likely increase the rates of consumption of natural food sources, whether that be planktonic sources or plant material growing in the ponds. As seen here (Fig 5), tilapia in T2 (alternate day feeding along with pond fertilization) had the greatest diversity of eukaryotic organisms identified in their guts. This could have led to a greater variety of nutrients available to them for growth than the tilapia in the other treatment groups.

Alpha diversity measures are indices of the diversity within a community. The Chao 1 index is commonly used to estimate the total number of species within a community and is based on the number of rare OTUs in that community [71]. The diversity of prokaryotes was significantly higher in Treatments 1 (fed daily with pond fertilization) and 2 (alternate day feeding with pond fertilization) than in the other three treatments while T2 had the highest diversity of eukaryotes and Treatments 3 (fed every third day with pond fertilization) and 5 (daily feeding only) had the lowest. It is predicted that ecosystems with high species-richness better utilize limiting resources [72–73] and are more resilient to disease states such as inflammatory bowel disease [74] and *Clostridium difficile*-associated pathologies [75]. Thus, increased diversity in the gut microbiome could enhance health, survival, and feed efficiency in farmed tilapia. As T5 (daily feeding only) also received daily feeding but had significantly lower Chao 1 indices relative to T1 (fed daily with pond fertilization), it is likely that pond fertilization alone is beneficial for growing tilapia as it increases the availability of natural food sources providing a more varied diet. Alternate day feeding (T2) appears to also be beneficial for enhancing microbe diversity as these fish are likely consuming a greater proportion of plankton and plant materials than those fed daily (T1).

## Conclusion

It is estimated that 50–70% of total variable costs for growing tilapia is attributable to formulated feeds. This study demonstrated that alternate-day feeding reduces the costs of feeds for growout of tilapia by 50%, increases feed efficiency by almost 76%, and has little impact on growth, survival or yield of tilapia farmed in ponds. The results also suggest that tilapia grown in fertilized ponds without supplemental feeds may be nutritionally impaired as key nutrient transporters in the gut are enhanced in preparation for increased uptake of solutes should food become available, a process that is mitigated when animals are provided supplemental feeds. The intermediate expression of gut nutrient transporters in alternate-day fed tilapia may reflect a condition for the most efficient uptake of nutrients from the GI tract. Finally, this work shows that the use of fertilizers and the implementation of an alternate-day feeding strategy increase the diversity of intestinal microbiota that may function in promoting nutrient assimilation and in these fish. These microbes may serve as promising candidates for isolation and development of probiotics beneficial to increased feed efficiency, growth, and health in tilapia.

## Supporting information

**S1 Fig. Chao1 rarefaction plots of microbial communities identified from the anterior intestine of Nile tilapia subjected to reduced-feeding regimes (mean ± SEM).** (a) 16S rRNA (prokaryotes); (b) 18S rRNA (eurkaryotes).
(TIF)

**S1 Table. Primers used in studies for Nile tilapia (*O. niloticus*) subjected to reduced-feeding regimens.**
(XLSX)

**S2 Table. Water quality parameters of ponds used in studies for Nile tilapia (*O. niloticus*) subjected to reduced-feeding regimens.** Values are mean ± SD. Values with different letters are significantly different ($P < 0.05$).
(XLSX)

**S3 Table. Plankton abundance in ponds used in studies for Nile tilapia (*O. niloticus*) subjected to reduced-feeding regimens.** Values are mean ± SEM. Values with different letters are significantly different ($P < 0.05$).
(XLSX)

**S4 Table. Proportions by treatment of prokaryotes and eukaryotes identified to the family level in fecal material from Nile tilapia subjected to reduced-feeding regimens.**
(XLSX)

**S5 Table. Unique prokaryotic Operational Taxonomic Units (OTUs) identified in the fecal material from Nile tilapia subjected to reduced-feeding regimens.**
(XLSX)

# Acknowledgments

The authors would like to thank Dr. Hillary Egna of Oregon State University for her leadership of the Aquafish Innovation Lab and her intellectual contributions to the work. The opinions expressed herein are those of the authors and do not necessarily reflect those of the AquaFish Innovation Lab-USAID or the NSF.

# Author Contributions

**Conceptualization:** Md. Abdul Wahab, David A. Baltzegar, Russell J. Borski.

**Data curation:** Scott A. Salger.

**Formal analysis:** Scott A. Salger.

**Funding acquisition:** Md. Abdul Wahab, Russell J. Borski.

**Investigation:** Jimi Reza, Md. Abdul Wahab, Alexander T. Murr, Russell J. Borski.

**Methodology:** Scott A. Salger, David A. Baltzegar, Alexander T. Murr.

**Project administration:** Md. Abdul Wahab, Russell J. Borski.

**Resources:** Md. Abdul Wahab, Russell J. Borski.

**Supervision:** Md. Abdul Wahab, Russell J. Borski.

**Validation:** Scott A. Salger.

**Visualization:** Scott A. Salger.

**Writing – original draft:** Scott A. Salger.

**Writing – review & editing:** Scott A. Salger, Courtney A. Deck, Russell J. Borski.

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
