## [Decision Letter · Decision Letter 0]

24 Feb 2020

PONE-D-19-35983

Enhanced biodiversity of gut flora and feed efficiency in pond cultured tilapia under reduced frequency feeding strategies

PLOS ONE

Dear Dr. Salger,

Thank you for submitting your manuscript to PLOS ONE. After careful consideration, we feel that it has merit but does not fully meet PLOS ONE’s publication criteria as it currently stands. Therefore, we invite you to submit a revised version of the manuscript that addresses the points raised during the review process.

Two reviewers have evaluated your manuscript, found the data compelling and of interest, but it does require some revisions.   Most of the revisions recommended are editorial in nature and should be straight forward.  However, there are some suggestions that may take a bit more thought; although there are no recommendations for further experiments.

We would appreciate receiving your revised manuscript by May 15, 2020.. To enhance the reproducibility of your results, we recommend that if applicable you deposit your laboratory protocols in protocols.io, where a protocol can be assigned its own identifier (DOI) such that it can be cited independently in the future. For instructions see: http://journals.plos.org/plosone/s/submission-guidelines#loc-laboratory-protocols

We look forward to receiving your revised manuscript.

Kind regards,

Michael H. Kogut, Ph.D.

Academic Editor

PLOS ONE

Journal Requirements:

Reviewers' comments:

Reviewer's Responses to Questions

**Comments to the Author**

1. Is the manuscript technically sound, and do the data support the conclusions?

Reviewer #1: Yes

Reviewer #2: Yes

2. Has the statistical analysis been performed appropriately and rigorously? 

Reviewer #1: Yes

Reviewer #2: I Don't Know

3. Have the authors made all data underlying the findings in their manuscript fully available?

Reviewer #1: Yes

Reviewer #2: Yes

4. Is the manuscript presented in an intelligible fashion and written in standard English?

Reviewer #1: Yes

Reviewer #2: Yes

5. Review Comments to the Author

Reviewer #1: The paper is very well written and provides strong data to support the theories proposed. Description of specific hypotheses and acceptance or rejection would have improved the paper. A few minor editorial comments are attached to the review copy.

The work here explains a phenomenon noted by many farmers. Description of the gene expression levels of intestinal nutrient transporters along with the characterization of the gut microbial community provide a strong case of why we observe the benefit of alternate day feeding.

This is novel research in aquaculture and could provide the scientific basis for farmers who have been leery to adapt the practice.

Reviewer #2: The objective of this paper was to evaluate the underlying mechanisms for improved feed efficiency of Nile tilapia when fed on alternate days. In particular, nutrient transporter gene expression and fecal microbiota were assessed. The authors conclude that that alternate-day feeding reduces the costs of feeds for grow out of tilapia by 50%, increases feed efficiency by almost 76%, and had little impact on growth, survival or yield of tilapia farmed in ponds. This paper is well written and easy to read. Please address the following comments:

1. The authors need to re-run their analysis. Pond is the experimental unit as treatment was applied to the 4 ponds out of the 20.

2. More discussion or analysis would be useful in relating FCR differences to mRNA and microbial data. This is lost in looking at the data just be treatment.

3. Line 46: need to reference statistic.

4. Line 67: poorly understood in fish. Livestock has data on this.

5. Line 69: Reference needed.

6. Line 71-72: What has been postulated?

7. Line 192: a more accurate anatomical location of sampling needs to be described. How were the samples handled between euthanasia and RNA extraction?

8. Table 1: Split the economic analysis into a new table. Further, the economic assumptions need to be stated. How did you get to the USD?

9. Table 1: Why was T4 FCR not reported?

10. Line 328: P-value to 3 decimals.

11. Line 340-346: Please refer to mRNA as abundance not levels.

12. Line 391: P-0.255 not a trend.

13. Line 511: Don’t reference Figures in Discussion.

14. Line 519: What causes if not different.

15. Figure 1: Was there a week by treatment interaction?

16. Figure 2: Is the y-axis fold change?

17. Line 253: Is pond or fish the experimental unit. It should be pond. Please state. What was the significance level set at?

18. Please add treatments to figures and not just Treatment 1 etc… It will be easier to follow your figure.

6. PLOS authors have the option to publish the peer review history of their article (what does this mean?). If published, this will include your full peer review and any attached files.

Reviewer #1: Yes: Kevin Fitzsimmons

Reviewer #2: No

---

## [Author Response · Author response to Decision Letter 0]

29 Jun 2020

Reviewer #1: Manuscript Number: PONE-D-19-35983

The paper is very well written and provides strong data to support the theories proposed. Description of specific hypotheses and acceptance or rejection would have improved the paper. A few minor editorial comments are attached to the review copy.

The work here explains a phenomenon noted by many farmers. Description of the gene expression levels of intestinal nutrient transporters along with the characterization of the gut microbial community provide a strong case of why we observe the benefit of alternate day feeding.

This is novel research in aquaculture and could provide the scientific basis for farmers who have been leery to adapt the practice.

Thank you for your comments provided within the manuscript. We have addressed them in the revised manuscript. 

Reviewer #2: Manuscript Number: PONE-D-19-35983

Reviewer #2: The objective of this paper was to evaluate the underlying mechanisms for improved feed efficiency of Nile tilapia when fed on alternate days. In particular, nutrient transporter gene expression and fecal microbiota were assessed. The authors conclude that that alternate-day feeding reduces the costs of feeds for grow out of tilapia by 50%, increases feed efficiency by almost 76%, and had little impact on growth, survival or yield of tilapia farmed in ponds. This paper is well written and easy to read. Please address the following comments:

1. The authors need to re-run their analysis. Pond is the experimental unit as treatment was applied to the 4 ponds out of the 20. 

Yes, the pond was our experimental unit and our statistical analysis was performed using this information. For all statistical tests performed, each sample within the pond was pooled followed by the appropriate test being run. We have edited the methods section to make these details clearer (lines 143-156). Thank you for alerting us to this issue.

2. More discussion or analysis would be useful in relating FCR differences to mRNA and microbial data. This is lost in looking at the data just be treatment.

We performed correlation tests between FCR and gene expression of each solute transporter or microbiome diversity measures but found no significant correlations between FCR and these measures. The relatively small number of samples (ponds), the fact that FCR is an estimate based on the amount of commercial feed applied to the pond, and the complexity with which FCR is controlled (treatment type, commercial diet, natural pond production, nutrient availability, individual variation, transporters not evaluated here, etc.), does not allow us to make definitive conclusions other than data suggesting alternate-day feeding along with pond fertilization leads to intermediate expression of solute transporter genes and an increased level of prokaryotic and eukaryotic diversity in the fecal material of these fish. We postulate that together these factors along with the higher biodiversity of organisms available for consumption (commercial diet and natural pond productivity) may contribute to greater feed efficiency (lower FCR). We have added a statement of this in the discussion, lines 461-467. 

3. Line 46: need to reference statistic.

We have added the FAO The State of World Fisheries and Aquaculture 2020 document as reference to this while also updating the information with the most recent data (see lines 45-46). The latest information from FAO indicates that the share of aquaculture has increased to greater than 40% worldwide. Thank you.

4. Line 67: poorly understood in fish. Livestock has data on this.

We have clarified this statement to include a discussion of only finfish and not all livestock (line 65-67).

5. Line 69: Reference needed.

We have added references to these statements (lines 69). Thank you.

6. Line 71-72: What has been postulated?

This statement has been modified for clarity: “This increase in nutrient uptake efficiency following periods of decreased nutrient uptake has been postulated, in part, to explain the compensatory growth (CG) response observed in many aquaculture species [7-11].” (lines 70-73)

7. Line 192: a more accurate anatomical location of sampling needs to be described. How were the samples handled between euthanasia and RNA extraction? 

Thank you for alerting us to this lack of clarity. We have described the location of the sampling for both gene expression and rRNA studies in the Fish and Pond Sampling section of the Materials and Methods (lines 143-145) and added reference to these in their analysis sections (lines 194-196). The samples were stored in RNALater at room temperature during shipment and immediately stored at -80°C until extraction. This was clarified in the Fish and Pond Sampling section (lines 145-147). 

8. Table 1: Split the economic analysis into a new table. Further, the economic assumptions need to be stated. How did you get to the USD? 

We agree that splitting this table into two aids in the quality of the table, thus we now include a Table 1 detailing the growth and production performance of the Nile tilapia used in this study and Table 2 detailing the economic analyses that were run. Along with this, all adjustments were made in order to properly cite these tables. We also added a statement on the economic assumptions of this study to the Results section (lines 309-311): “The economic assumption of alternate-day feeding is that it would lead to more cost effective tilapia aquaculture production, which could enhance income of small scale farmers and consumption of nutrient rich fish.” Last, we provide information under Table 2 of the revised manuscript on how US dollar (USD) values were attained, along with additional information on other operational costs (labor, leasing, etc.) which were excluded from calculations. Were labor included in our cost-benefit analysis, alternate-day feeding would lead to more economic savings above that associated with the current analysis. 

9. Table 1: Why was T4 FCR not reported? 

Thank you for this question. Since no commercial diet was offered to the fish in this treatment group (pond fertilization only) FCR could not be calculated. Therefore, FCR values were not reported for this group. 

10. Line 328: P-value to 3 decimals.

This adjustment has been made. 

11. Line 340-346: Please refer to mRNA as abundance not levels.

This adjustment has been made.

12. Line 391: P-0.255 not a trend.

We have removed reference to this being a trend and clarified this statement: “Although there was no significant difference in species richness due to sample variation within treatments (P = 0.256), treatments with greater access or variety of foodstuff (feed and fertilization) had higher diversity measures.” (lines 404-406)

13. Line 511: Don’t reference Figures in Discussion. 

While it is not essential that figures be cited in the discussion, we feel that periodically referencing figures and tables when discussing the data may help the reader in understanding the statements being made. Therefore, we wish to keep the references to figures/tables in the manuscript. 

14. Line 519: What causes if not different. 

Thank you for this comment. We have updated this statement as it was not necessary for the argument being made. We do discuss the solute transporters studied here later in the referenced paragraph and the following paragraph. Here, we discuss the nutrients transported and the mechanisms in which gene expression was affected by the potential availability of the specific nutrients (line 533-534).

15. Figure 1: Was there a week by treatment interaction? 

Although, we only report the final difference in Figure 1, there was a significant wek (time) by treatment interaction (P < 0.001). For length (figure 1a), there was no difference between fish only fed a prepared diet every day or fish fed the diet everyday along with pond fertilization throughout the trial. There was a significant difference between these two treatments and the fish fed on alternate days in fertilized ponds weeks 7 through 13, but no significant difference between these three treatments at weeks 15 and 17. Weight (figure 1b) followed similar trends, except that there was also no significant difference in these three treatments as weeks 13, 15, and 17. We have reflected this interaction in the revised manuscript (lines 283-290). 

16. Figure 2: Is the y-axis fold change? 

Thank you for this oversite. No, we calculated our gene expression using cDNA copy number standard curves. These measures do not indicate fold change. These indicate copy number normalized to 18S rRNA.

17. Line 253: Is pond or fish the experimental unit. It should be pond. Please state. What was the significance level set at? 

Thank you for this question. Yes, the ponds were our experimental unit in all of the statistical analysis performed in this study. Significance was assigned at an α of 0.05. (line 275)

18. Please add treatments to figures and not just Treatment 1 etc… It will be easier to follow your figure.

We thank the reviewer for this comment. We agree that by adding the treatments to the figures and tables it aids in their comprehension. The treatment names, along with the treatment numbers, have been added to all figures and tables, including the supplementary files.

---

## [Editor Report · Decision Letter 1]

30 Jun 2020

Enhanced biodiversity of gut flora and feed efficiency in pond cultured tilapia under reduced frequency feeding strategies

PONE-D-19-35983R1

Dear Dr. Salger,

We’re pleased to inform you that your manuscript has been judged scientifically suitable for publication and will be formally accepted for publication once it meets all outstanding technical requirements.

Kind regards,

Michael H. Kogut, Ph.D.

Academic Editor

PLOS ONE
---

## [Editor Report · Acceptance letter]

6 Jul 2020

PONE-D-19-35983R1 

Enhanced biodiversity of gut flora and feed efficiency in pond cultured tilapia under reduced frequency feeding strategies 

Dear Dr. Salger:

I'm pleased to inform you that your manuscript has been deemed suitable for publication in PLOS ONE. Congratulations! Your manuscript is now with our production department. 

Kind regards, 

on behalf of

Dr. Michael H. Kogut 

Academic Editor

PLOS ONE